

# Response of soil pH to biochar application in farmland across China: a meta-analysis

Jia Yao[1,2], Xueren Wang[1,2], Mei Hong[3], Hui Gao[1,2] and Shixiang Zhao[1,2]

[1] Key Laboratory of Agricultural Ecological Security and Green Development at Universities of Inner Mongolia Autonomous Region, Hohhot, China
[2] Inner Mongolia Agricultural University, Inner Mongolia Key Laboratory of Soil Quality and Nutrient Resource, College of Resources and Environment, Hohhot, China
[3] College of Grassland, Inner Mongolia Minzu University, Tongliao, China

## ABSTRACT

**Background:** Biochar, an alkaline material derived from agricultural and forestry waste, can ameliorate soil quality by adjusting soil pH. However, various types of biochar have distinct effects on soil pH due to diversity in feedstock type, pyrolysis temperature, and application rate.

**Method:** Therefore, a meta-analysis was conducted with 598 paired comparisons obtained from 104 published studies (January 2010–July 2022) to comprehensively depict the response of soil pH to biochar in farmland systems across China.

**Result:** The results showed that adding biochar significantly increased the acid soils' pH. Still, its effects on neutral and alkaline soils varied depending on the biochar's pH and the soil's initial pH. The pH of acid and neutral soils was raised by 5–10% straw biochar (600–800 °C and 400–600 °C, respectively). In alkaline soils, 5–10% other biochar (200–400 °C) raised pH, while 1–5% straw biochar (200–400 °C) reduced it. The findings underscore the importance of biochar pH and initial soil pH in the divergent consequences of biochar application in farmland systems, and both factors should be considered to choose the optimal biochar type for acid, neutral, and alkaline soils.

## INTRODUCTION

Soil pH is a key index to evaluate the quality of soil fertility and plays an important role in biogeochemical cycle. It is closely related to the level of soil fertility, the activity of microorganisms and fauna, and the formation of humus, and has a direct impact on the form and effectiveness of soil nutrients (*Zeng et al., 2011*; *Chen et al., 2019*; *Li et al., 2019*). To tackle soil acidification or alkalisation problems and boost crop productivity, increasing effort has been directed toward ameliorating soil pH in farmland systems.

Biochar is a carbon-rich material prepared by high-temperature pyrolysis of biomass under fully or partially anoxic conditions. Common feedstocks include agricultural waste (*e.g.*, crop straw, woody peat) (*Demirbas, 2006*), livestock manure, and domestic waste (*e.g.*, sewage sludge, municipal solid waste) (*Gunarathne et al., 2019*). Biochar is generally alkaline, with weak acidity in a small proportion. It is characterized by rich pore structure,

Corresponding author
Shixiang Zhao,
zhaoshixiang1989@126.com

large specific surface area, small specific gravity, stable physicochemical properties, and abundant surface functional groups (*Hossain et al., 2020*; *Liang et al., 2021*). Owing to these characteristics, the addition of biochar confers benefits to soil structure and fertility. However, the effects of biochar on soil quality are variable depending on the feedstock, temperature, and method used for biochar preparation (*Zhang, Voroney & Price, 2015*; *Zhao, Ta & Wang, 2017*).

For instance, the application of woody biochar at different pyrolysis temperatures (450 °C and 550 °C) and application rates (0%, 3%, 6%, and 10%) to acidified sandy soil (pH = 3.8) in central Portugal significantly increased soil pH to a range of 5.0–7.0 (*Morim et al., 2023*). Similarly, biochar prepared from a mixture of sludge and rice husk applied to Central European silty loam (pH = 5.6) increased soil pH by 0.4–0.9 units (*Hematimatin et al., 2024*). In the Czech Republic, biochar application to acidic (pH = 4.3) and neutral (pH = 6.8) loam soils resulted in significant pH increases, with neutral soils showing a pH increase of 1.2–1.3 units (*Száková et al., 2024*; *Mikajlo et al., 2024*).

Irrespective of the type of biochar applied, it has the potential to raise the pH level of acid soils. The pH-adjusting capability of biochar in acid soils has been reported for sludge biochar (*Zong et al., 2018*; *Malik et al., 2018*), straw biochar (*Yu et al., 2017*; *Wu et al., 2018*), and woody biochar (*Pan et al., 2021*; *Yan et al., 2021*). However, different types of biochar exhibit divergent effects on soil alkalinity (*Qiang et al., 2020*; *Hu et al., 2021*; *Liu, Xie & Zhang, 2017*; *Elkhlifi et al., 2021*). Specifically, biochar application can lead to three different outcomes in alkaline soils, *i.e.*, no noticeable change in soil pH (*Song et al., 2019*, *2020*), decrease in soil pH (*Zhao et al., 2020*), and increase in soil pH (*Cui et al., 2022*). Despite a plethora of research highlighting the effects of biochar on soil pH, uncertainties remain on the influence of distinct feedstocks, temperatures, supplements, and their interactions. Consequently, it is imperative to devise guidelines for biochar application that account for varying soil pH.

Previous meta-analysis has elucidate the effects of biochar on greenhouse gas emissions (*Cayuela et al., 2014*; *Jeffery et al., 2016*; *He et al., 2017*), plant root traits (*Xiang et al., 2017*; *Zou et al., 2021*), heavy metal bioavailability (*Tian et al., 2021*; *Yuan et al., 2021*), and crop yield (*Jeffery et al., 2011*; *Crane-Droesch et al., 2013*). Despite some meta-analyses revealing biochar-induced changes in soil physicochemical properties (*Sun et al., 2022*), there is still a paucity of research deciphering the responses of soil pH to various types of biochar in farmland systems across large spatial scales. To fill this knowledge gap, we compared the effects of biochar with different feedstock types, pyrolysis temperatures, and application rates on the pH of acid, neutral, alkaline soils in through a meta-analysis of published articles. Results of this study could provide mechanistic insights into the divergent role of biochar in improving soil quality from a pH perspective.

# MATERIALS AND METHODS

## Data sources

The Web of Science (https://www.webofscience.com/) and CNKI (https://www.cnki.net/) databases were systematically searched for relevant studies published from January 2010 to

July 2022. The keywords used for the search were (biochar OR black carbon OR charcoal) AND (soil pH). Jia Yao and Shixiang Zhao evaluated the title and abstract of each study to determine whether it contained original data on the response variable used in the present study. In case of disagreement, seek the help of other co-authors to determine the primary referee by vote. The selection criteria were as follows: (1) the object of the study was farmland soil in China; (2) the design of the study was field experiment, plot experiment, or pot experiment; (3) the experiment consisted of at least one treatment with biochar and one treatment without biochar (control), and other field conditions were consistent; (4) the basic physicochemical properties of soil and biochar samples were reported; and (5) each treatment was replicated at least three times. Exclusion criteria: (1) The research object does not match: excluding the study of non-farmland soil (such as forest soil, grassland soil, *etc.*); (2) Research design does not match: excluding laboratory simulation tests, model simulation studies or the lack of control group studies; (3) Incomplete data: Excluding studies that did not report the basic physical and chemical properties of soil or biochar, or studies with incomplete data that could not be effectively analyzed; (4) Insufficient repetition: studies with less than three repetitions per treatment were excluded.

A total of 104 valid research articles were retrieved and 598 sets of data were collected. The flow chart of literature removal is shown in Fig. 1. The province and latitude and longitude of the soil sampling points for each test are listed in File S2. We recorded the observed data of the response variable—soil pH—in each article, including the mean and standard deviation (SD) or standard error (SE) of control and biochar treatments. In addition to directly extracting tabular data, we digitized figure data using GetData version 2.25 (Getdata Pty Ltd, Manhattan Beach, CA, USA). Moreover, the data of the original soil pH and biochar characteristics (feedstock, pyrolysis temperature, application rate, pH) were recorded. All data were standardized appropriately before further analysis. The biochar application rate was converted into a unified unit (%). If SEs were reported in a selected article, they was converted to SDs using Eq. (1). For articles that did not report SDs or SEs, SDs were estimated to be 10% of the means (*Luo, Hui & Zhang, 2006*).

$$SD = SE\sqrt{n}. \tag{1}$$

The data were categorized by soil pH level and biochar characteristics as follows. Based on their original pH, soils were divided into three groups: acid (pH < 6.5), neutral (pH 6.5 = 7.5), and alkaline (pH > 7.5). Biochar feedstocks were also classified into three types: crop straw (*e.g.*, wheat husk, rice straw, corn cob), woody matter (*e.g.*, timbers, leaves, branches), and others (*e.g.*, sewage sludge, livestock and poultry manure, municipal solid waste). *Li & Chen (2018)* found that the physical and chemical properties of high temperature biochar ≥600 °C and ≤400 °C biochar are different, so according the biomass pyrolysis temperatures were ranked into three levels: low (200–400 °C), medium (400–600 °C), and high (600–800 °C). Biochar application rates were grouped into four levels: low (<1%), medium (1–5%), high (5–10%), and extremely high (10–20%).
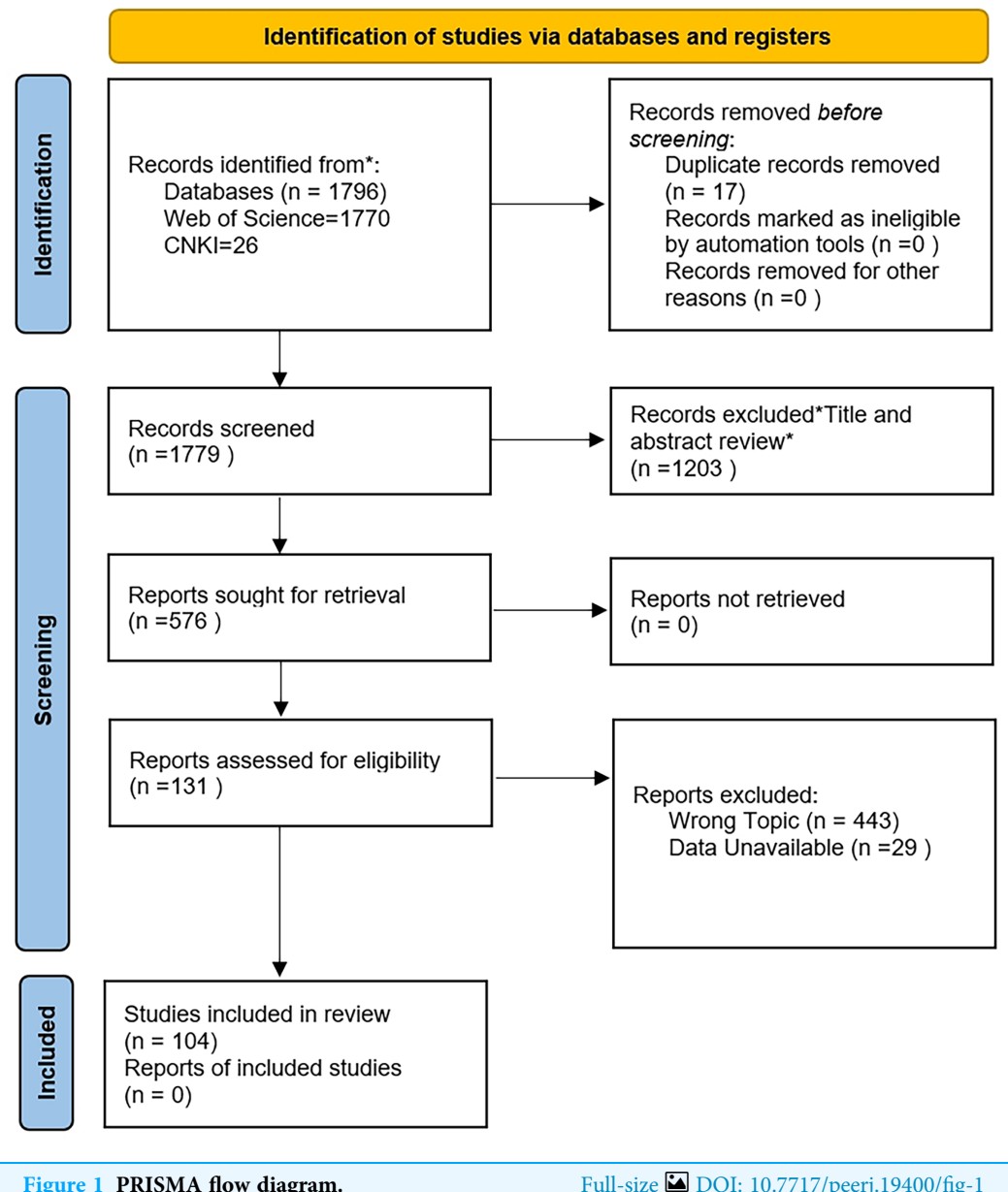

**Figure 1  PRISMA flow diagram.**

## Data analysis

In the meta-analysis, the treatments with and without biochar addition were defined as the treatment and control groups, respectively. We computed effect sizes as natural log-transformed response ratios (lnR):

$$lnR = \ln\left(\frac{X_t}{X_c}\right) = \ln X_t - \ln X_c \tag{2}$$

where $X_t$ and $X_c$ are the mean responses of soil pH in the treatment and control groups, respectively. The variance ($v$) of response ratio was estimated as follows:

$$v = \frac{S_t^2}{n_t X_t^2} + \frac{S_c^2}{n_c X_c^2} \tag{3}$$

where $n_t$ and $n_c$ are the sample sizes for the treatment and control groups, respectively; $S_t$ and $S_c$ are the SD values for the treatment and control groups, respectively.

We conducted weighted meta-analysis, where the mean effect size of each categorical variable was calculated using a fixed effects model. Groups with less than two treatments were omitted from the analysis. Metawin version 2.0 (*Rosenberg, Adams & Gurevitch, 2000*) was adopted to calculate the overall mean effect size and 95% confidence interval (CI) for each categorical variable. If the 95% CI did not overlap with zero, the mean effect size was considered significant; $X_t$ and $X_c$ were considered significantly different if the 95% CI did not overlap from one another. One-way analysis of variance with Tukey's test was used to determine whether there were significant differences in soil pH among different groups or levels of categorical variables. Publication bias is one of the most concerned and studied bias in Meta-analysis. In order to test the publication bias of the collected literature and ensure the accuracy of the evaluation results and conclusions of meta-analysis, this study uses Eggar's method to quantify. When the bias test value $P > 0.05$, it is considered that there is no publication bias in the collected literature, and vice versa (*Egger et al., 1997*). Statistical analyses were conducted using OriginPro version 2022 (OriginLab Corp., Northampton, MA, USA).

## RESULTS

### Total effect of biochar on soil pH

Averaged across the entire dataset, biochar addition significantly increased soil pH in the samples, with an effect value of 0.0450 (Fig. 2). When separating the samples into acidic, neutral, and alkaline soils, biochar exhibited a positive effect on the pH of acidic soils (0.0662), and this effect was greater than that for neutral (0.0136) and alkaline soils (0.0054).

### Effect of biochar from different feedstocks on soil pH

The addition of biochar derived from various feedstocks displayed a positive effect on soil pH (Fig. 3). The response of soil pH to straw biochar was greater in acid soils (0.0801) than in neutral and alkaline soils (0.0135 and 0.0033, respectively). Woody biochar exhibited a similar effect to straw biochar in both acid and alkaline soils. However, other biochar displayed a greater effect in neutral soils, with no significant difference between acid and alkaline soils. Among the three feedstock types, woody biochar and straw biochar had prominent effects on acid soils (0.0820 and 0.0801, respectively), which accounted for a significant increase in soil pH. On the contrary, other biochar had the greatest effect on neutral (0.0498) and alkaline (0.0441) soils.

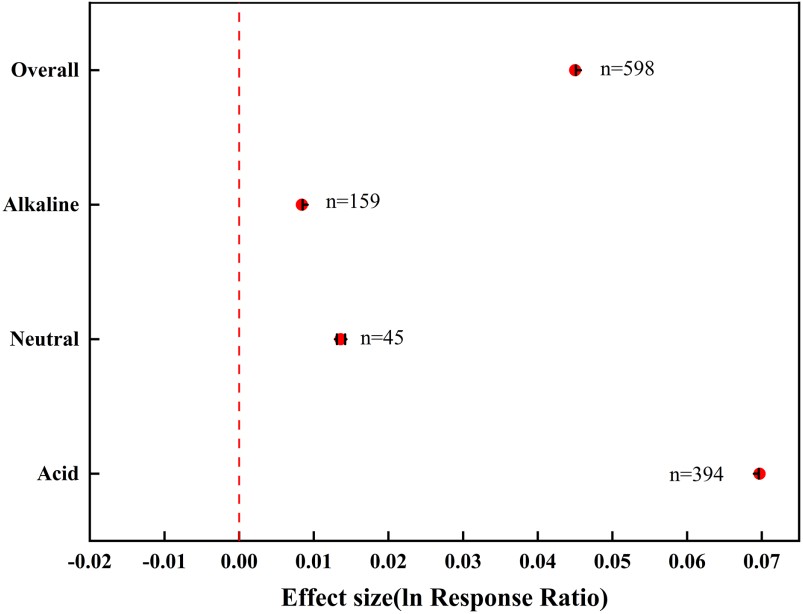

**Figure 2** **The total effect of biochar on soil pH in farmland systems.** The effect size was considered statistically significant if the 95% bootstrap confidence interval (CI) did not include zero. The numbers next to the bars are sample sizes for each variable (*n*). 

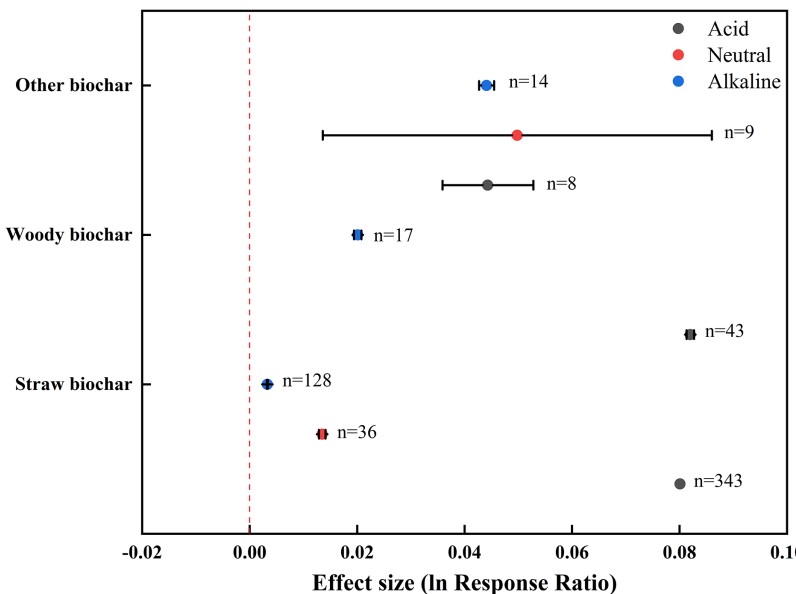

**Figure 3** **Effect of biochar from different feedstocks on soil pH in farmland systems.** The effect size was considered statistically significant if the 95% bootstrap confidence interval (CI) did not include zero. The numbers next to the bars are sample sizes for each variable (*n*).

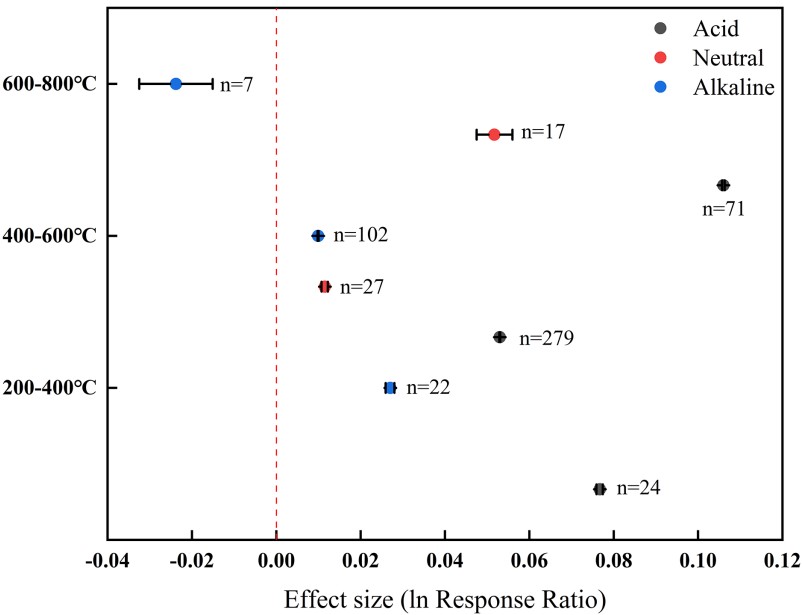

**Figure 4** **Effect of biochar with different pyrolysis temperatures on soil pH in farmland systems.** The effect size was considered statistically significant if the 95% bootstrap confidence interval (CI) did not include zero. The numbers next to the bars are sample sizes for each variable ($n$).

## Effect of biochar with different pyrolysis temperatures on soil pH

Upon addition of biochar prepared at 200–400 °C, the pH of both acid and alkaline soils responded positively, and the effect on acid soils was greater than that on alkaline soils (Fig. 4). The addition of biochar prepared at 400–600 °C exhibited a positive effect on soil pH in all three groups, and the greatest effect emerged in acid soils. Notably, adding the biochar prepared at 600–800 °C exerted a negative effect on the pH of alkaline soils (−0.0238), which contrasted with its positive effect on the pH of neutral and acid soils. Among the three temperature levels, the positive effect of high-temperature biochar was the most remarkable on acid (0.1008) and neutral (0.0517) soils. However, for alkaline soils, the greatest negative effect was attributed to high-temperature biochar (−0.014), whereas the most evident positive effect was posed by low-temperature biochar (0.0270).

## Effect of biochar applied at different rates on soil pH

The effects of different biochar application rates on soil pH were divergent (Fig. 5). Biochar addition at low rates <1% mainly positively affected the pH of acid (0.0319) and neutral (0.0223) soils, with a minimal negative effect on alkaline soils (−0.0017). The addition of 1–5% and 5–10% biochar had an exclusively positive effect on soil pH over a range of acidity, neutrality, and alkalinity. The positive effect of biochar was generally enhanced with increasing application rate, and adding 10–20% biochar specifically affected the pH of acid and neutral soils. Among the three levels of application rate, 5–10% biochar exhibited

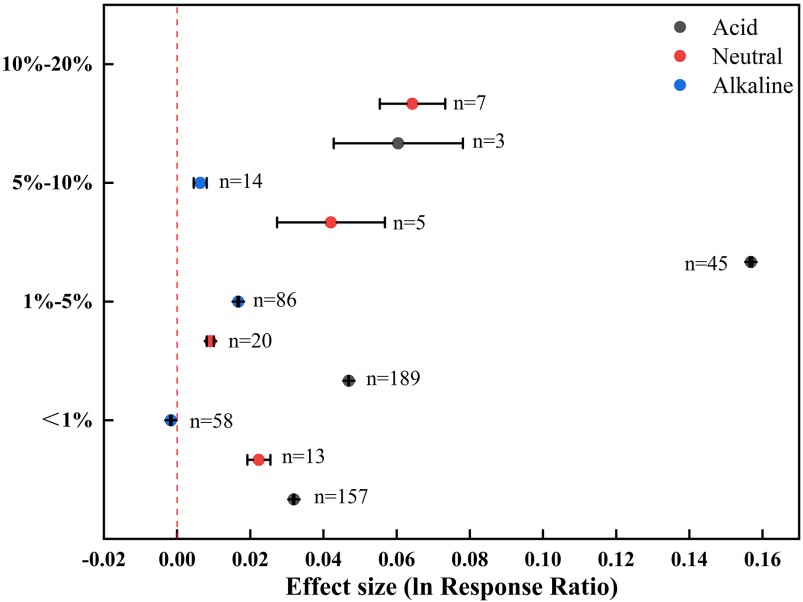

**Figure 5 Effect of different biochar application rates on soil pH in farmland systems.** The effect size was considered statistically significant if the 95% bootstrap confidence interval (CI) did not include zero. The numbers next to the bars are sample sizes for each variable (*n*).

the greatest positive effect on acid soils (0.1568), whereas 10–20% and 1–5% biochar respectively showed the most prominent effect on neutral (0.0643) and alkaline (0.0167) soils.

## Interaction effects of biochar feedstock and pyrolysis temperature on soil pH

Biochar feedstock and pyrolysis temperature showed varied interaction effects on soil pH (Fig. 6). The responses of pH in acid and neutral soils to straw biochar with different pyrolysis temperatures (Fig. 6A) were consistent with the response patterns to all biochar (Fig. 4). In the case of alkaline soils, straw biochar prepared at 200–400 °C and 400–600 °C had a slight positive effect on soil pH (0.0003 and 0.0075, respectively), whereas a negative effect was observed for straw biochar prepared at 600–800 °C (−0.0250).

The positive effect of woody biochar on the pH of acid soils was heightened with increasing pyrolysis temperature. In alkaline soils, woody biochar showed contrasting effects on soil pH depending on pyrolysis temperature, as indicated by a positive effect value at 400–600 °C (0.0210) and a negative effect value at 600–800 °C (−0.0238; Fig. 6B). The effects of other biochar prepared at different temperatures were positive across all soil groups (Fig. 6C).

## Interaction effect of biochar feedstock and application rate on soil pH

There were distinct interaction effects of biochar feedstock and application rate on soil pH (Fig. 7). Specifically, different straw biochar rates had a positive effect on the pH of acid and neutral soils (Fig. 7A), and the response of pH in acid soils was consistent with the

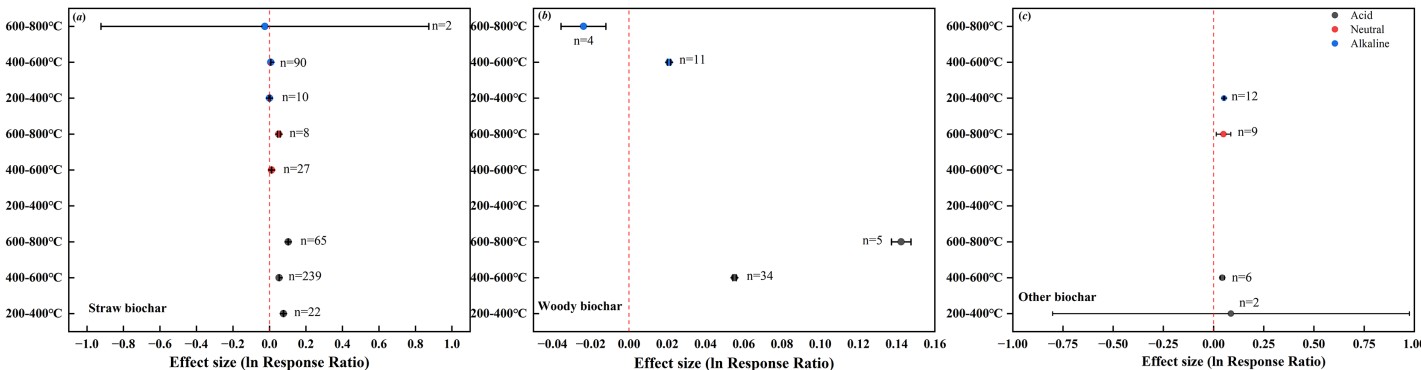

**Figure 6 Interaction effects of biochar feedstock and pyrolysis temperature on soil pH in farmland systems.** The effect size was considered statistically significant if the 95% bootstrap confidence interval (CI) did not include zero. The numbers next to the bars are sample sizes for each variable (*n*).

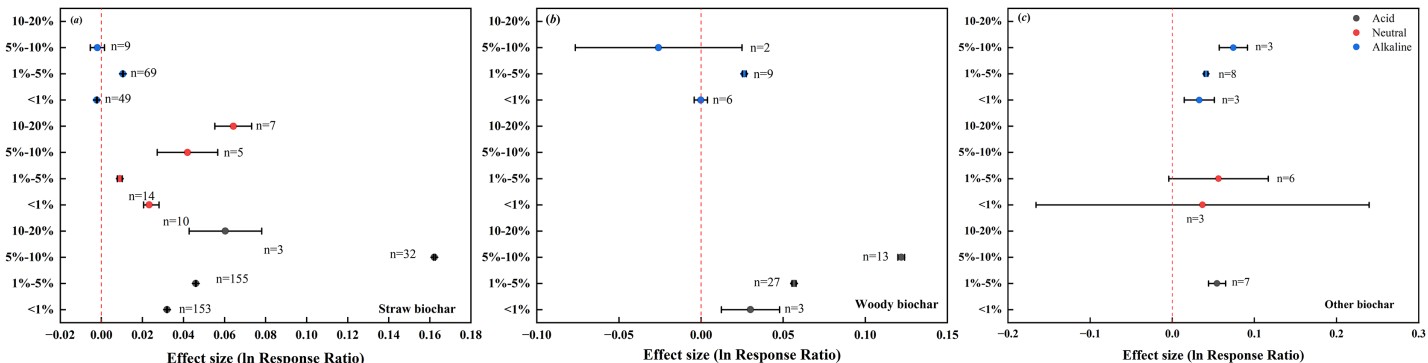

**Figure 7 Interaction effects of biochar feedstock and application rate on soil pH in farmland systems.** The effect size was considered statistically significant if the 95% bootstrap confidence interval (CI) did not include zero. The numbers next to the bars are sample sizes for each variable (*n*).

pattern observed for all biochar (Fig. 5). In the case of alkaline soils, adding 1–5% straw biochar had a small positive effect (0.0105) on soil pH, whereas lower and higher straw biochar rates had a minor negative effect on soil pH (–0.0022 and –0.0019, respectively; Fig. 7A).

The response of pH in acid soils to different woody biochar rates was also consistent with the pattern observed for all biochar (Fig. 5). In alkaline soils, the effect of woody biochar rate on soil pH resembled that of straw biochar rate. When woody biochar was added at rates of 1–5%, there was a positive effect on soil pH (0.0263). Conversely, a negative effect emerged with woody biochar addition at lower and higher rates (–0.0001 and –0.0259, respectively; Fig. 7B).

The addition of other biochar at different rates had consistent positive effects on the pH of acid, neutral, and alkaline soils (Fig. 7C). Together, the results indicate that greatest positive effect on soil pH was observed in acid and neutral soils upon straw biochar addition at high (5–10%) and extremely high (10–20%) rates, respectively. The greatest negative effect observed in alkaline soils was attributed to woody biochar addition at high rates.

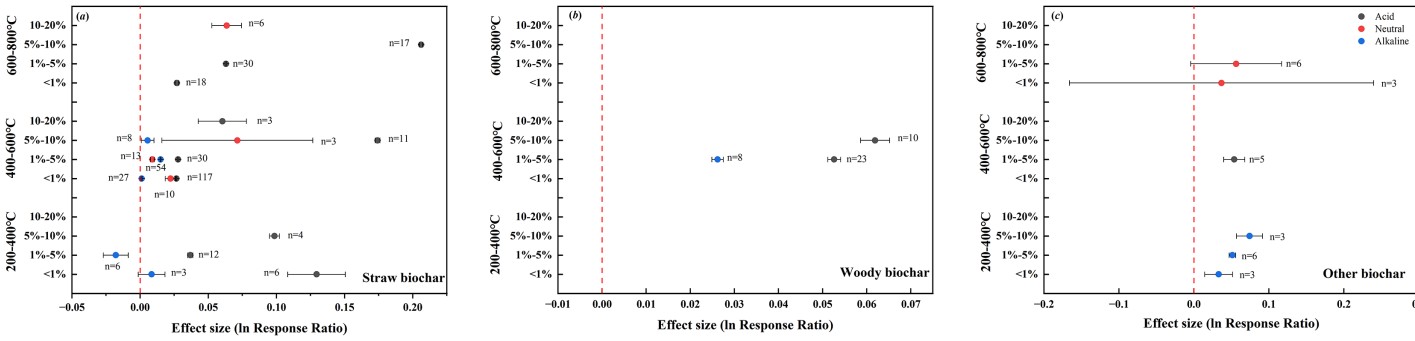

**Figure 8 Interaction effects of biochar feedstock, pyrolysis temperature, and application rate on soil pH in farmland systems.** The effect size was considered statistically significant if the 95% bootstrap confidence interval (CI) did not include zero. The numbers next to the bars are sample sizes for each variable (*n*).

## Interaction effect of biochar feedstock, pyrolysis temperature, and application rate on soil pH

Regardless of the application rate, the addition of straw biochar prepared at 200–400 °C exhibited a positive effect on the pH of acid soils (0.0767). However, when this biochar was applied between rates of 1–5%, a negative effect on soil pH emerged in alkaline soils (−0.0178). The effect of straw biochar prepared at 600–800 °C was positive across all soils, being the strongest in acid soils at the application rate of 5–10%. Woody biochar prepared at 400–600 °C also positively affected the pH of both acid and alkaline soils, and this effect was strengthened in acid soils with increasing application rate. The addition of other biochar with different pyrolysis temperatures had a positive effect on the pH of acid, neutral, and alkaline soils, and there was an upward trend in this effect with increasing application rate (Fig. 8C).

Among the different treatments, the pH of acid soils was positively affected the most by addition of 5–10% straw biochar with high temperatures (600–800 °C; 0.2062). The same application rate of medium-temperature straw biochar (400–600 °C) was most effective in increasing the pH of neutral soils (0.0713). In alkaline soils, however, the greatest positive response of pH was observed upon addition of 5–10% other biochar prepared at 200–400 °C (0.0742), whereas the addition of 1–5% straw biochar prepared at 200–400 °C had a negative effect on alkaline soils (−0.0178).

## DISCUSSION

Large-scale application of biochar, a promising soil amendment, is plagued by divergent consequences in different soils. In this meta-analysis, we demonstrated the benefits of biochar addition in ameliorating the pH of acid, neutral, and alkaline soils in farmland systems across China based on experimental data published over the past 12 years (Fig. 2). More importantly, we linked biochar-induced changes in soil pH to feedstock type, pyrolysis temperature, and application rate (Figs. 3–5). The findings present a holistic picture of how soil pH responds to various types of biochar in farmland systems, which has practical implications for the development of guidelines on biochar application.

The addition of biochar with various feedstocks, pyrolysis temperatures, and application rates notably increased the pH of acid farmland soils (*Shi et al., 2019*) (Figs. 3–8). This positive effect on soil pH is attributable to the alkaline nature of biochar. During pyrolysis, the acid functional groups and cations present in the feedstock are incorporated to form alkaline substances, such as –COO–, –O–, carbonates, and oxides. After biochar application, these alkaline substances play a role in alleviating soil acidity by neutralizing $H^+$ ions and reacting with $Al^{3+}$ ions in the soil (*Dai et al., 2017*). *Fuertes et al. (2010)* showed that 1 ton of hydrochar could have a limiting effect of 39.6 kg $CaCO_3$. In addition to $H^+$ and $Al^{3+}$ ion exchange, biochar can elevate soil pH through the input of cations, such as $Ca^{2+}$, $K^+$, and $Mg^{2+}$ (*Glaser, Lehmann & Zech, 2002*). Therefore, it is not surprising that the pH of acidic soils increases in response to biochar addition.

Our analysis revealed feedstock-dependent variations in pH modulation efficiency. In acid soils, woody biochar exhibited a greater effect than straw biochar and other biochar, whereas in neutral and alkaline soils, other biochar exerted the greatest effect on soil pH (Fig. 3). According to *Wijitkosum (2022)* and *Schmidt & Wilson (2012)*, biochar made from woody matter has low pH as a result of low ash content. As such, woody biochar may not be as effective in increasing the pH of acid soils as biochar derived from crop straw and other materials. This contradicts our conclusion based on the meta-analysis. One possible reason for this contradiction is that in our dataset, the pH ranges of straw biochar (6.48–11.32) and other biochar (7.97–8.80) collected from acid soils are lower than that of woody biochar (7.90–11.30). Therefore, woody biochar has the best performance in increasing the pH of acid soils. In alkaline soils, the pH of other biochar is generally higher than soil pH, whereas woody biochar and straw biochar sometimes have lower pH values lower than the soil. This explains why other biochar has an overall stronger positive effect on the pH of alkaline soils. The situation of neutral soils is similar to that of alkaline soils.

Furthermore, pyrolysis temperature emerged as a critical determinant of biochar's pH-modifying capacity. High-temperature biochar was more effective in improving the pH of acid soils compared with low-temperature biochar (Fig. 3), consistent with the findings of *Sani et al. (2020)*. Carbonization at higher temperatures can enhance the dehydration and decomposition of organic acids, generating more alkaline substances in biochar (*Geng et al., 2022*). *Wan et al. (2014)* applied biochar with three different pyrolysis temperatures (300 °C, 500 °C, 700 °C) to improve the pH of acid soil. They observed a greater increase in soil pH under the application of biochar produced at 700 °C, which had a higher alkalinity, as compared with biochar produced at lower temperatures. *Guo & Rockstraw (2007)* indicated that the number of acid functional groups in activated carbon decreased with increasing pyrolysis temperature. The greatest decrease in acid functional groups emerged at 300–400 °C, and the loss rate of these functional groups slowed down at >400 °C.

Our results showed that high-temperature biochar was more effective in increasing the pH of acid and neutral soils in farmland, whereas low-temperature biochar had a greater effect on alkaline soils (Fig. 4). *Liu & Zhang (2012)* reported that alkaline biochar did not increase the pH of five types of alkaline soils, but instead produced a decreasing pH trend. The increase in the carboxylic and phenol functional groups due to natural oxidation of
biochar added in soil may be the possible reason for the decline in the pH (*Yadav et al., 2019*). Furthermore, during the oxidation on biochar surface, oxygen may be chemisorbed to unsaturated ring point of biochar carbon, leading to a subsequent formation of oxygenated functional groups (*Wang, Xiong & Kuzyakov, 2016*). This pattern was consistent with regard to feedstock type, such as straw, wood, or other materials (Fig. 6). It is noteworthy that the data collected in our dataset were mainly related to the effects of different application rates of straw biochar on soil pH, with limited data available for woody biochar and other biochar (Fig. 8). *Iyobe et al. (2004)* have suggested that the decomposition of lignin and cellulose within the temperature range of 400–500 °C is responsible for the decrease in biochar pH. Our data also indicate that the pH of biochar produced at 600–800 °C (7.25–11.32) is generally lower, albeit not significantly, compared with that of biochar produced at 400–600 °C (7.10–11.30). However, soils treated with biochar produced at 600–800 °C generally showed higher initial pH values (4.31–8.97) than soils treated with biochar produced at other temperatures (3.78–8.59). Consequently, low-temperature biochar exhibited a stronger effect in elevating soil pH, and the effect of biochar pH being lower than the initial soil pH was diminished.

As the application rate of biochar increased, the pH in farmland soils responded positively (Figs. 5, 7, 8), in agreement with previous studies (*Molnár et al., 2016*; *Laird et al., 2017*). *Zhou et al. (2019)* showed that the pH values of acid soil were elevated with increasing rate of biochar application in the range of 1–4%, and the pH value of 4% biochar-treated soils was 0.11 units higher than that of the control. *Liang et al. (2014)* also showed that soil pH value increased with increasing biochar addition from 30 to 90 t·hm$^{-2}$, despite the effect was indistinct. Biochar has the ability to neutralize soil acidity, partially because it contains $Ca^{2+}$, $Mg^{2+}$, $K^+$, and $Na^+$ ions in the form of oxides and soluble carbonates, which can dissolve in water and form alkalies (*Tan et al., 2017*).

The results reported in this study showed that the addition of biochar with different feedstocks and pyrolysis temperatures had similar effects on the pH of acid and neutral soils, which contrasted with the response of pH in alkaline soils (Figs. 4, 6). In principle, the higher the pyrolysis temperature, the higher the biochar pH (*Geng et al., 2022*). However, the pH of alkaline soils responded differently to woody biochar and straw biochar, with higher pyrolysis temperatures contributing less to soil pH. This is because based on our data, the soils treated with biochar prepared at higher temperatures had higher initial pH values. The responses of soil pH to different application rates of woody biochar and other biochar were inconsistent with the response patterns to the application rates of total biochar in alkaline soils. This was also due to the minimal difference between biochar pH and soil pH in our dataset.

## Limitations

SE to SD depends on the accuracy of the sample size *n*. If the original study does not report *n* or there is a selective report, it may lead to SD being overestimated or underestimated, thus distorting the calculation of effect size. Assume that SD = 10% mean will be divorced from reality. SD reflects the inherent dispersion of data and has no fixed proportional relationship with the mean (Fig. 9). Therefore, using these two methods to estimate SD

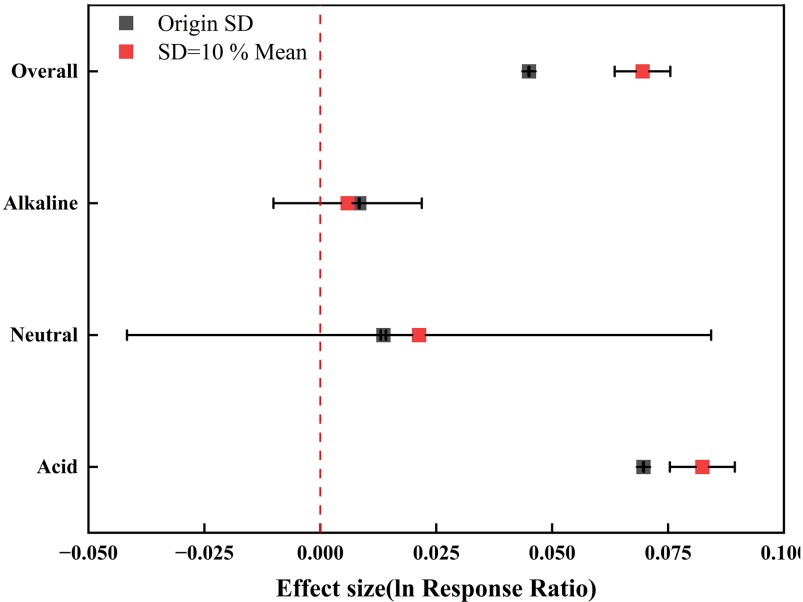

**Figure 9 The effect value visualization of different standard deviation processing methods.** Assuming that SD = 10% Mean, the difference of acid soil effect value is less than 20%, but the difference of neutral and alkaline soil is 56.61% and 29.76%, respectively, indicating that neutral and alkaline soil results are more sensitive to SD hypothesis.

may cause errors in the results. In the future research, the original data should be obtained first, and the advanced statistical methods such as maximum likelihood estimation should be used to deal with the missing SD, and the influence direction of different SD assumptions on the results should be tested in the sensitivity analysis.

The data of this study focused on short-term experiments (<3 years), which could not reflect the long-term effect of biochar aging (such as oxidation, microbial degradation) on pH (*Lehmann et al., 2021*); the regulation of soil microbial communities (acid-producing bacteria and nitrifying bacteria) on biochar effect was not quantified. The sample size in some areas (alkaline soil in the arid region of Northwest China) was insufficient (File S2), which may affect the universality of the conclusion. It is suggested that future research should combine long-term positioning experiments and metagenomic techniques to reveal the interaction mechanism of biochar-microorganism-soil chemistry.

## CONCLUSIONS

Based on the meta-analysis, this study systematically depicted the divergent responses of soil pH to various types of biochar in farmland systems, using Chinese farmland soil as a case in point. The results indicated that biochar addition generally increased the pH of farmland soils, despite the effects varying with feedstock type, pyrolysis temperature, and application rate. In both acid and neutral soils, the addition of woody biochar, other biochar, and high-temperature pyrolysis (600–800 °C) notably increased soil pH. In the case of straw biochar, an extremely high application rate (10–20%) was effective in elevating the pH of neutral soils, whereas a high rate (5–10%) was optimal for acid soils.

Furthermore, the pH of alkaline soils was mainly increased upon addition of low-temperature other biochar (200–400 °C), medium-temperature woody biochar (400–600 °C), and high-rate woody biochar (5–10%), which contrasted with the effect of high-temperature biochar. When ameliorating acid, neutral, and alkaline soils, it is crucial to select the optimal biochar type by taking into account the biochar pH and initial soil pH. Therefore, it is recommended to apply 5–10% straw biochar at 600–800 °C and 400–600 °C in acidic and neutral soils, respectively, to maximize soil pH, and to apply 1–5% straw biochar in alkaline soils to reduce soil pH.

### Funding

This work was funded by the National Key R&D program (2022YFD1500901-03) and the National Natural Science Foundation of China (41907082). The funders had no role in study design, data collection and analysis, decision to publish, or preparation of the manuscript.

### Grant Disclosures

The following grant information was disclosed by the authors:
National Key R&D program: 2022YFD1500901-03.
National Natural Science Foundation of China: 41907082.

### Competing Interests

The authors declare that they have no competing interests.

### Author Contributions

- Jia Yao conceived and designed the experiments, performed the experiments, analyzed the data, prepared figures and/or tables, and approved the final draft.
- Xueren Wang analyzed the data, prepared figures and/or tables, and approved the final draft.
- Mei Hong conceived and designed the experiments, prepared figures and/or tables, authored or reviewed drafts of the article, and approved the final draft.
- Hui Gao conceived and designed the experiments, authored or reviewed drafts of the article, and approved the final draft.
- Shixiang Zhao conceived and designed the experiments, performed the experiments, analyzed the data, authored or reviewed drafts of the article, and approved the final draft.

### Data Availability

The raw data is available in the Supplemental File.

### Supplemental Information

Supplemental information for this article can be found online at http://dx.doi.org/10.7717/peerj.19400#supplemental-information.

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
