# Peer review of "Response of soil pH to biochar application in farmland across China: a meta-analysis"

_PeerJ, doi:10.7717/peerj.19400_

## Round 0.1 · original submission · Major Revisions

Dear authors, I kindly ask you to revise the manuscript very carefully in accordance with all the recommendations of the reviewers. I hope that the new version of this article will be approved by the reviewers and will be published in our journal as soon as possible.

Reviewer 1 ·

Basic reporting

attached

Experimental design

attached

Validity of the findings

attached

Annotated reviews are not available for download in order to protect the identity of reviewers who chose to remain anonymous.

Reviewer 2 ·

Basic reporting

Language and Expression: The manuscript is generally well-written, but some sentences are lengthy or grammatically complex.
Literature Review: While the literature review is extensive, it lacks critical analysis of the most recent studies. Consider adding key research from 2022-2024 relevant to soil pH regulation to enhance the completeness of the background.

Experimental design

Data Sources and Selection Criteria: The data screening criteria are briefly mentioned but do not detail the specific reasons for excluding studies. Provide more comprehensive inclusion and exclusion criteria to improve transparency.
Grouping Logic: The grouping of soil pH and biochar characteristics is scientifically sound, but the boundaries between "high temperature" (600–800°C) and "medium temperature" (400–600°C) are not well justified. Elaborate on the rationale for these classifications.

Validity of the findings

Depth of Results: Although the results are detailed, the analysis related to neutral and alkaline soils is relatively brief. For example, the data interpretation in Figure 5 is shallow. Provide a deeper mechanistic analysis to explain the differences in biochar effects on these soil types.
Statistical Limitations: The manuscript does not mention potential biases due to small sample sizes or large intergroup differences. Include a discussion on the statistical limitations and present sensitivity analysis results.
Practical Applications: While the discussion emphasizes theoretical implications, it lacks concrete recommendations for agricultural practices (e.g., suggested application rates and frequencies). Provide actionable advice tailored to farmland conditions in China or other countries.

---

## Round 0.2 · Minor Revisions

Dear Dr. Yao, I ask you to make minimal changes to the manuscript and send me the final version for publication consideration.

Reviewer 1 ·

Basic reporting

no comment

Experimental design

no comment

Validity of the findings

no comment

Additional comments

Suggest accept and publish

Reviewer 2 ·

Basic reporting

This study addresses an important scientific question with significant academic and practical implications. The manuscript is well-structured, logically sound, and methodologically rigorous. The experimental design is appropriate, the data analysis is thorough, and the conclusions are well-supported. The authors have carefully addressed the previous review comments, making substantial improvements to the manuscript, resulting in a more precise and rigorous presentation.

Experimental design

The study is well-designed with appropriate variable control, scientific data collection methods, and compliance with established research standards in the field.

Validity of the findings

The study presents novel insights that contribute valuable knowledge to the field, demonstrating both academic significance and practical applicability.

Additional comments

Quality of Revisions: The authors have carefully and thoroughly addressed all comments, improving the manuscript in terms of completeness, clarity, and logical coherence.

·

Basic reporting

The article is aimed at solving an important problem of efficient agricultural production and improving soil quality through the use of biochar. The problem is really important and interesting, and much has already been done in this direction. The authors need to more clearly define the main result of their work, so that it differs significantly from the already known data and information.

Further recommendations could probably help improve the quality of the article.

Line 36: “Soil pH is a fundamental chemical property that impacts terrestrial biogeochemical processes” – A dubious statement. A property cannot influence processes. According to Dokuchaev, the sequence is different: factors→processes→properties.

Line 37: “Excessive acidification or alkalization of the soil reduces nutrient availability” – The category of ‘’excessive‘’ cannot be applied to soil: soil has no measure for this characteristic. But plants have.

Line 38: “It also hinders soil structure formation” – acidification or alkalisation have completely different mechanisms of influence on the soil structure, so they cannot be combined.

Line 39 “seriously inhibiting microbial activity“ – acidification or alkalisation have completely different mechanisms of action on different groups of soil microorganisms, so they cannot be combined.

Line 39 “To tackle pH-related problems …“ – The authors did not disclose the nature of the problem, since the deviation of pH from the optimum for plants (probably meaning agricultural plants) is only the tip of the iceberg.

Line 71: The sentence is not related to the topic of the article and is more appropriate to the research methods.

Experimental design

Materials & Methods are well presented, relaxed and provide reliable results. The results of the study are presented in a complete and meaningful manner.

In discussing the results of a meta-analysis of the literature, it is not appropriate to present the material in a discussion style such as ‘Our results corroborated earlier research’, since all the authors' results are the result of previous research by other authors.

Conclusions: «Based on the meta-analysis, this study systematically depicted the divergent responses of soil pH to various types of biochar in farmland systems across China» – The authors are restricted to farmland systems across China for the purposes of clarity. This may limit the interest of a wide range of readers, so the authors should identify a general problem, the solution of which, using the example of China, will be of interest to a wide range of readers.

Validity of the findings

The well-organised nature of the study and the large number of studies used for the meta-analysis make its results reliable. The results have scientific and practical significance. Nevertheless, the need to identify an important result, which was established by the authors, is an important condition for the publication of this article.

---

## Round 0.3 · accepted · Accept

Dear Dr. Yao,

I congratulate you on the acceptance of your article for publication in our journal. I hope that you will continue this research in the future and I look forward to your next articles of the same high quality.

·

Basic reporting

The authors have implemented all the recommendations of the reviewer. The quality of the manuscript has been significantly improved. I recommend the article for publication

Experimental design

The authors have implemented all the recommendations of the reviewer. The quality of the manuscript has been significantly improved. I recommend the article for publication

Validity of the findings

The authors have implemented all the recommendations of the reviewer. The quality of the manuscript has been significantly improved. I recommend the article for publication

Additional comments

The authors have implemented all the recommendations of the reviewer. The quality of the manuscript has been significantly improved. I recommend the article for publication